# Who can go back to work when the COVID-19 pandemic remits?

**Luis Angel Hierro[1], David Cantarero[2], David Patiño[1]\*, Daniel Rodríguez-Pérez de Arenaza[1]**

**1** Department of Economics and Economic History, University of Seville, Seville, Spain, **2** Department of Economics, Group of Health Economics and Health Services Management, University of Cantabria-IDIVAL, Santander, Spain

\* pato@us.es

## Abstract

This paper seeks to determine which workers affected by lockdown measures can return to work when a government decides to apply lockdown exit strategies. This system, which we call Sequential Selective Multidimensional Decision (SSMD), involves deciding sequentially, by geographical areas, sectors of activity, age groups and immunity, which workers can return to work at a given time according to the epidemiological criteria of the country as well as that of a group of reference countries, used as a benchmark, that have suffered a lower level of lockdown de-escalation strategies. We apply SSMD to Spain, based on affiliation to the Social Security system prior to the COVID-19 pandemic, and conclude that 98.37% of the population could be affected. The proposed system makes it possible to accurately identify the target population for serological IgG antibody tests in the work field, as well as those affected by special income replacement measures due to lockdown being maintained over a longer period.

**Data Availability Statement:** The data underlying the results presented in the study are uploaded to Zenodo and publicly accessible via the following URL: https://zenodo.org/record/3971821#. Xyra8igzaF5.

## 1. Introduction

On December 31, 2019, the first case of COVID-19 was reported by China to the WHO [1]. On January 30, 2020, the WHO declared a "Public Health Emergency of International Concern (PHEIC)" [2] and on March 11, 2020 declared that COVID-19 could be considered a pandemic, with cases in 114 countries [3]. On January 23, Chinese authorities reacted to mass contagion in the city of Wuhan by imposing major restrictions on the mobility of people, in other words a lockdown, which was later extended by varying degrees to other parts of the area affected [4]. This measure was subsequently applied in Italy, initially only to certain northern regions and later to the country as a whole. Finally, Spain directly applied a national lockdown, without exceptions (Royal Decree 463/2020, March 14). Since then, many of the countries affected have been applying lockdown measures, although with very different characteristics and to varying degrees and normally adopting less stringent lockdown measures than those imposed in Spain.

A full-scale nationwide lockdown is a drastic and controversial measure, which affects both social and economic activities. The benefit it produces has an immediate impact on health.

**Funding:** The authors received no specific funding for this work.

Mortality is drastically reduced by cutting viral transmission routes, and thereby preventing health systems from collapsing [5]. Furthermore, it allows for a lower incidence of the disease while an effective vaccine or treatment appears. In contrast to this health benefit, there are very high costs, especially in terms of the economy. Lockdown reduces GDP and increases the number of unemployed [6]. It can also send stock market prices [7] tumbling by dissipating financial wealth, added to which it jeopardises the liquidity of many people and entities [8] and with it that of financial institutions, . . .

In other words, lockdown saves lives, but paralyzes the economy and can trigger an economic shock of enormous proportions [9]. This extremely high economic cost makes deciding when to lockdown a difficult issue for public authorities. Similarly, lifting lockdown restrictions is also a difficult decision since, if contagion reappears, the country or region will have borne a very high economic cost without the measures having proved effective in reducing the effects of the pandemic. Obviously, these decisions are even more difficult to take when the country is one of those most affected, since it is not known how effective the measures being taken have been or what the likelihood is of causing a resurgence of the virus.

As the reproductive rate of the virus has declined, the WHO and the countries affected have begun to consider criteria for gradually bringing an end to the lockdown. The WHO establishes public health criteria for lockdown de-escalation strategies, while it is the governments who must combine these criteria with economic and political considerations in order to decide who lockdown exit strategy should be applied to as well as when and how.

In decentralized countries, it is common to see proposals for geographical lockdown exit strategies. Such is the case of the United States, where public health powers are in the hands of state governors, or Spain, where health care is under the control of the autonomous communities (regions) while public health is run by the central government. Whatever the case, in addition to the political circumstances, selective or asymmetric geographical lockdown exit strategies are fully understandable in large countries which have substantial epidemiological differences.

As regards the economic approaches aimed at gradually bringing lockdown to an end, which is the case in hand, the most obvious criterion is the sectoral criterion. In most countries, lockdown has been selective by sectors of activity depending on the social contact involved. It is also reasonable, therefore, to propose that lockdown exit strategies should follow the same pattern. In the case of COVID-19, it will not be possible for the lockdown exit strategy to be a symmetrical but reverse process to the lockdown strategy since, while there is no vaccine, many services will be forced to change their production organization and it will not be possible to meet demand in the same way as before the COVID-19 pandemic. However, proposing a selective lockdown de-escalation strategy by activity sectors based on how the virus has been transmitted is inevitable, since this is how lockdown was applied [10].

Furthermore, empirical evidence shows that mortality is directly related to patient age. Older people tend to suffer from the infection in its most virulent form and many end up requiring admission to intensive care units or even dying. This means that some countries promote deconfinement strategies designed to protect the most vulnerable age groups. In the United Kingdom, the proposal for age-selective lockdown exit strategies has been in place since the beginning, and there are now proposals for age-selective lockdown exit strategies [11].

Finally, there is the proposal for a lockdown exit strategy by immunity detected through serological tests. The proposal involves deconfining all of those who test positive for antibodies by issuing an immunity passport [12,13]. For this to prove feasible in the workplace, all workers would need to be tested. However, the WHO opposes this type of proposal and has issued a scientific note stating that *"There is currently no evidence that people who have recovered from*

*COVID-19 and have antibodies are protected from a second infection"* [14]. As in the previous case, lack of immunity can lead to ethical and social justice issues [15].

Since there are different possible dimensions for lockdown exit strategies, in this document we present a system for deciding on the number of workers that could return to work according to said dimensions. This system involves defining a sequence of decisions to determine which working population lockdown exit strategy should be applied to by geographical area, sector of activity, age range, and immunity. The aim of applying Sequential Selective Multidimensional Decisioning (SSMD) is therefore to decide the number of workers who can return to work at a given time. It is important to emphasize that it does not seek to establish a timeframe for a lockdown de-escalation strategy, although it may be applied at successive moments to define a lockdown exit strategy timeline.

As an example, we apply the SSMD designed for Spain, whose characteristics in terms of lockdown and lockdown de-escalation strategies are summarized in S1 Appendix. As a reference, we use data on workers affiliated to the social security system in February 2020, at the start of the COVID-19 pandemic, and as health criteria we use contagion and mortality rates. The structure of the SSMD is flexible and can be adapted to different health criteria and to different levels of geographical, sectoral and age structure disaggregation.

The work is structured as follows. In section 2, we describe the method to be applied. In section 3, we include the description of the data used. Section 4 presents the results, and is followed with a discussion thereof in section 5. Finally, some brief conclusions are provided in section 6.

## 2. Methodology

### A.- Dimensions for the lockdown de-escalation strategy

Given the economic cost of lockdown, from the moment the virus' reproductive rate drops below one, authorities are faced with the decision of what lockdown de-escalation strategy to adopt. Authorities can either follow the previous course and completely reopen all economic-social activity, maintaining lockdown in its original state until final deconfinement, or apply a selective lockdown exit strategy in stages. The first option has the social and economic benefit of allowing all workers to return to their jobs, which avoids a deeper recession. However, it has the disadvantage of possible loss of health and human life if there is a fresh outbreak. The second, the other extreme, has the disadvantage of the loss of the social and economic benefit associated with economic paralysis, triggering a longer-lasting recession, although it would prevent the virus from re-emerging and thus prevent fatalities. In between the two lie all the selective or partial lockdown de-escalation strategies based on defining criteria related to relevant economic or social aspects over a longer or shorter period during which a gradual lockdown exit strategy is applied. The latter seeks to combine health benefits and reduced economic costs, and thus strike a balance.

The features of lockdown exit strategies can differ according to the type of virus, although there is always the possibility of a selective or asymmetric lockdown de-escalation strategy depending on:

- Geographical criteria: carrying out a selective or asymmetric geographical lockdown exit strategy, when there are substantial differences in the incidence of the epidemic.

- Sectors of activity: deconfining by sectors of activity depending on the contagion potential of each type of activity.

- Age: by deconfining citizens according to age, given that the disease does not impact the different age groups equally.

- Immunity: by deconfining all citizens who have tested positive through serological tests and are immune.

Public authorities can establish a one-dimensional lockdown exit strategy, using only one of these, or a multi-dimensional strategy by combining several of them.

For the lockdown process, Spain used the activity dimension sector, and applied it throughout the country as a whole: in other words, it used a one-dimensional lockdown strategy. For its lockdown exit strategy, the Spanish government has developed a selective multidimensional strategy based on distinguishing by areas and by sector.

## B.- The decision on how many workers may return to work

At each point in the lockdown de-escalation process, whether at a single moment or at the beginning of each phase, public authorities must decide on how many workers to apply the lockdown exit strategy to. This paper seeks to answer this question. To this end, we propose adopting a selective multidimensional approach, using the four criteria for the above-mentioned lockdown exit strategy: geographical, sector of activity, age, and immunity.

The process we apply is described in the decision tree in Fig 1.

As shown in the figure above, the process consists of an orderly sequence of selective lockdown exit strategy decisions.

- **Decision Level I.** First, we decide on the selective geographical lockdown exit strategy, considering the minimum territorial division for which we have data on all the variables involved.

When health authorities analyse geographical scope, the main concern is to control the rate of infection. Based on this, for the first decision level, we divided areas into two groups: low and high COVID-19 cumulative infection rate. To define this benchmark level, as a reference group we use countries with a below average rate of lockdown exit strategies in the European Union (plus the UK), assessed from the data on variation in mobility provided by Google and we calculate the average cumulative infection rate for the group.

We classify as low mortality areas those below the average European incidence value, and as high mortality areas those above said value, and we apply the full-scale geographical lockdown exit strategy to areas with a low mortality rate, while those with a high mortality move on to the second phase of decision.

This geographically selective lockdown exit strategy requires strong border control and enormous social discipline.

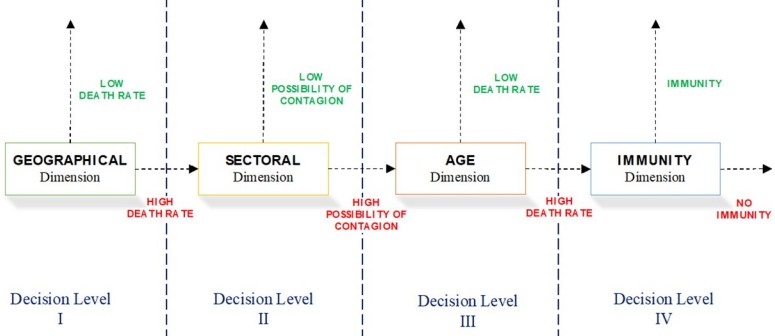

**Fig 1. Graphical description of the proposed Sequential Selective Multidimensional Decisioning (SSMD) process for post-pandemic lockdown de-escalation strategies by COVID-19.** Source: Own elaboration.

- **Decision Level II.** For high mortality areas, we apply a selective sectorial lockdown exit strategy criterion. We cannot establish any objective system in this regard since there is no information on the disease that would allow us to deduce an effective difference between sectors in terms of mortality. The only reference is that given by the lockdowns. Based on this information, we have divided the sectors by their capacity for contagion, considering the social relations involved and following the lockdown criteria generally employed by governments. The result is two groups of sectors of activity: sectors with high contagion possibility, if they are usually affected by lockdowns; and low contagion possibility, if they are usually less affected by lockdowns.

Applying this criterion, for the areas that have moved on to Decision Level II we fully deconfine sectors that display a low possibility of contagion while those with a high possibility of contagion pass to the third Decision Level.

The problem here concerns the difficulty involved in complying with social distancing in the workplace and the pressure being exerted by those sectors which are pushing for a lockdown exit strategy, given their weight in GDP. In this case, public authorities must be strict vis-à-vis establishing working conditions and the provision for demand that will protect the health of both workers and consumers alike.

- **Decision Level III.** For activity sectors in areas still under lockdown and that pass to this decision level, we apply a selective lockdown exit strategy by age. For the decision on a lockdown exit process, we again take as a reference the same group of European countries, but use the mortality rate by age cohort as data, since in this case what is relevant when deciding in individual health terms is the possibility of a subject dying due to the disease. We therefore use the average cumulative mortality rate per 100,000 inhabitants for the group. Thus, if the age group has a lower mortality rate than the European one, we apply a lockdown exit strategy to that age group, whereas if it does not, then the age group will move on to the next decision level. By applying the criterion to the age groups of the different area sectors of activity that have reached this stage, we can bring a significant proportion of workers in those sectors back to work. In other words, the idea is to apply lockdown exit strategies in accordance with the active population pyramid of the sectors of activity, and responds to the fact that the virus is not hitting all age groups in the same way.

In this case, protection of workers at the workplace is crucial, and public authorities must be even stricter when establishing working conditions and demand provision.

- **Decision Level IV.** The result of the previous decision level is that the most vulnerable age groups in the most vulnerable sectors remain in lockdown only in areas suffering the highest incidence of the virus. In this case, the serological criterion would be applied. These workers would be the first to undergo serological testing, such that if they have antibodies to SARS-CoV-2 they would be given an immunity passport and could return to work. Obviously, precise data on incidence will not be available until serological tests are carried out. To fill this gap, we return to the European reference and, for all available studies [16], take the serological study carried out in Germany on a community with a high incidence of the disease, Gangelt, which could resemble Spain, and which reports 14% seropositive [17].

Decision Levels III and IV raise ethical issues that force a debate on the matter and require an assessment of the compensation to be given to those who, because they are more vulnerable, are forced to remain in lockdown. This compensation is no longer an emergency compensation, and the amount must therefore be close to replacing the income lost during lockdown.

Once the calculations of lockdown exit strategies associated with the SSMD have been made, we calculate the related mortality. For each decision level $i$ and each area $k$, we estimate the death toll ($DeathToll_{ik}$). To do this, we calculate the number of liberated workers in each age group $j$ in that area ($LibWorkers_{ik}$), multiplying the percentage of that age group with respect to the total in the province ($\%Workers_{jk}$) by the mortality rate for that age group ($MortRateAgeGroup_j$) which, in the absence of provincial data, we have taken for the country as a whole. The total number of deaths in the province is obtained by adding up the four levels. To calculate the total number of deaths, the number of deaths in each province is added up ($DeathToll_k$). For decision level IV, we set a mortality rate of 0 because workers are immune.

We perform the calculation from expression (1):

$$DeathToll = \sum_{k=1}^{n} \sum_{i=I}^{IV} \sum_{j=16-29}^{60-69} LibWorkers_{i,k} \times \%WorkersAgeGroup_{j,k} \times MortRate_j$$

This allows us to calculate the mortality rate per 100,000 deconfined workers: that is, the mortality rate associated with the applied SSMD.

## 3. Data

The data on Spain and Europe that we use to apply the SSMD designed are as follows:

### A.- Social security affiliation data

The employment situation in Spain prior to the lockdown decision is obtained from the social security affiliation on February 29, 2020 [18]. We have calculated the total number of those affiliated in each activity group by adding up the corresponding number in all sections of the national social security system. In the case of the general section and the sections for self-employed workers, data are provided by the national social security system. Affiliates of the other special sections have been added to those activity groups most closely related to them. Specifically:

- Group A, for *agriculture, forestry and fishing*, also includes the special section for agricultural workers and the special section for seafarers, both those who are salaried and those who are self-employed.

- Group B, for *mining and quarrying*, also includes the special section for coal mining.

- Group T, for *household activities such as employers of domestic staff; household activities such as producers of goods and services for own use*, includes the special section for domestic staff.

    Data are included in S2 Appendix (Table A2.1 in S2 Appendix).
    For provincial distribution by age, we took provincial data from the general treasury of the social security [19] in average values of the same month. The distribution percentages are shown in Table A2.2 of S2 Appendix.

### B.- Epidemiological data on COVID-19 in Spain

We took the epidemiological data of COVID-19 in Spain for April 20, 2020. Table 1 includes provincialized data on cumulative infections and deaths, except for the autonomous communities of Catalonia and Galicia, for which no data are available. For these autonomous communities we chose to use aggregate data from them. The last two columns present the rates per 100,000 inhabitants. To prepare the table, we use data from the Spanish government website to

**Table 1. Accumulated COVID-19 infection and death rates per 100,000 inhabitants in Spain on April 20, 2020.** Data by province.

| AACC | Province | Population | Confirmed cases | Deaths | Conf. Cases / 100,000 hab | Deaths / 100,000 hab |
|---|---|---|---|---|---|---|
| ANDALUSIA | Almeria | 716,820 | 461 | 43 | 64.31 | 6.00 |
| | Cadiz | 1,240,155 | 1,146 | 75 | 92.41 | 6.05 |
| | Cordoba | 782,979 | 1,281 | 79 | 163.61 | 10.09 |
| | Granada | 914,678 | 2,078 | 205 | 227.18 | 22.41 |
| | Huelva | 521,870 | 389 | 34 | 74.54 | 6.52 |
| | Jaen | 633,564 | 1,309 | 140 | 206.61 | 22.10 |
| | Malaga | 1,661,785 | 2,546 | 223 | 153.21 | 13.42 |
| | Sevilla | 1,942,389 | 2,345 | 214 | 120.73 | 11.02 |
| ARAGON | Huesca | 220,461 | 601 | 80 | 272.61 | 36.29 |
| | Teruel | 134,137 | 541 | 65 | 403.32 | 48.46 |
| | Zaragoza | 964,693 | 3,678 | 491 | 381.26 | 50.90 |
| ASTURIAS | Asturias | 1,022,800 | 2,348 | 200 | 229.57 | 19.55 |
| BALEARIC, ISLANDS | Balearic, Islands | 1,149,460 | 1,788 | 157 | 155.55 | 13.66 |
| CANARY ISLANDS | Palmas, Las | 1,120,406 | 655 | 35 | 58.46 | 3.12 |
| | Santa Cruz de Tenerife | 1,032,983 | 1,430 | 86 | 138.43 | 8.33 |
| CANTABRIA | Cantabria | 581,078 | 2,083 | 158 | 358.47 | 27.19 |
| CASTILE—LA MANCHA | Albacete | 388,167 | 3,754 | 373 | 967.11 | 96.09 |
| | Ciudad Real | 495,761 | 6,358 | 802 | 1,282.47 | 161.77 |
| | Cuenca | 196,329 | 1,315 | 156 | 669.79 | 79.46 |
| | Guadalajara | 257,762 | 1,431 | 186 | 555.16 | 72.16 |
| | Toledo | 694,844 | 3,938 | 504 | 566.75 | 72.53 |
| CASTILE AND LEON | Avila | 356,958 | 1,567 | 168 | 438.99 | 47.06 |
| | Burgos | 460,001 | 2,403 | 303 | 522.39 | 65.87 |
| | Leon | 160,980 | 716 | 61 | 444.78 | 37.89 |
| | Palencia | 330,119 | 2,602 | 287 | 788.20 | 86.94 |
| | Salamanca | 153,129 | 2,406 | 172 | 1,571.22 | 112.32 |
| | Segovia | 88,636 | 1,243 | 96 | 1,402.36 | 108.31 |
| | Soria | 519,546 | 3,154 | 260 | 607.07 | 50.04 |
| | Valladolid | 172,539 | 611 | 65 | 354.12 | 37.67 |
| | Zamora | 157,640 | 1,155 | 109 | 732.68 | 69.14 |
| CATALONIA | CATALONIA | 7,675,217 | 43,802 | 4,247 | 570.69 | 55.33 |
| CEUTA | Ceuta | 84,777 | 111 | 4 | 130.93 | 4.72 |
| VALENCIAN COMMUNITY | Alicante | 1,858,683 | 3,577 | 401 | 192.45 | 21.57 |
| | Castellon | 579,962 | 1,325 | 144 | 228.46 | 24.83 |
| | Valencia | 2,565,124 | 5,437 | 539 | 211.96 | 21.01 |
| EXTREMADURA | Badajoz | 673,559 | 1,026 | 77 | 152.33 | 11.43 |
| | Caceres | 394,151 | 2,243 | 320 | 569.07 | 81.19 |
| GALICIA | GALICIA | 2,699,499 | 8,634 | 368 | 319.84 | 13.63 |
| MADRID, COMMUNITY OF | Madrid | 6,663,394 | 56,963 | 7,351 | 854.86 | 110.32 |
| MELILLA | Melilla | 86,487 | 104 | 2 | 120.25 | 2.31 |
| MURCIA, REGION OF | Murcia | 1,493,898 | 1,646 | 117 | 110.18 | 7.83 |
| NAVARRE | Navarra | 654,214 | 4,697 | 385 | 717.96 | 58.85 |
| BASQUE COUNTRY | Araba/Álava | 331,549 | 3,294 | 323 | 993.52 | 97.42 |
| | Gipuzkoa | 1,152,651 | 7,155 | 565 | 620.74 | 49.02 |
| | Vizcaya | 723,576 | 2,316 | 215 | 320.08 | 29.71 |
| RIOJA, LA | Rioja, La | 316,798 | 3,734 | 285 | 1,178.67 | 89.96 |

(Continued)

**Table 1.** (Continued)

| AACC | Province | Population | Confirmed cases | Deaths | Conf. Cases / 100,000 hab | Deaths / 100,000 hab |
|---|---|---|---|---|---|---|
| TOTAL | | 47,026,208 | 203,396 | 21,170 | 432.52 | 45.02 |

Source: Own elaboration based on Spanish government data [20], Escovid19data [21] and the National Institute of Statistics [22].

report on COVID-19 [20] and the Escovid19data [21] website which groups provincial data from various official sources.

The structure of the incidence of the disease by age in Spain is shown in Table 1, which is drawn up using the epidemiological data provided by the Carlos III Health Institute (Spanish acronym–ISCIII) [23]. The closest data to April 20 correspond to the report of April 21, 2020. The table includes data on infections and deaths accumulated by age groups (Table 2).

## C.- Calculation of the indicator of the level of deconfinement in Europe

In Europe, the level of lockdown has differed between countries. The approach adopted in this work is that if the geographical area of a country has similar data to those of countries with lower lockdown levels, then it can apply a lockdown de-escalation process to its workers. Differences in lockdown between countries are so high that it is impossible to make a comparative assessment. Google recently published mobility data based on the location of mobile phones as a measure of social distance in what they call COVID-19 Community Mobility Reports [24]. This database shows five measures of the degree to which individuals' mobility has changed during the health crisis. Specifically, it measures the change in travel to four different types of places: shops, recreation areas, restaurants, shopping malls or museums; grocery stores, supermarkets and pharmacies; national parks, beaches, public squares and gardens; transit stations, metros, buses or trains; workplaces. In general, the figures reflect a drop in the numbers. The database also measures the degree to which individuals have remained in their places of residence compared to the pre-pandemic situation.

From the published data, we have used the data referring to the latest available day as of writing this paper, April 17, 2020, to estimate an index of the degree of lockdown by taking the average of the absolute values of the percentage of reduction in movement and increased stay in places of residence offered by Google. We thus obtain an index that increases with the severity of the lockdown in each country. Table 3 shows this indicator, ranked from lowest to highest degree of lockdown.

**Table 2. Accumulated COVID-19 infection and death rates per 100,000 inhabitants in Spain on April 21, 2020.** Data by age groups.

| Age Group | Population | Confirmed cases | Deaths | Conf. cases / 100,000 hab | Deaths / 100,000 hab |
|---|---|---|---|---|---|
| From 0 to 4 | 2,029,628 | 325 | 2 | 16.0 | 0.1 |
| From 5 to 14 | 4,859,806 | 392 | 0 | 8.1 | 0.0 |
| From 15–29 | 7,212,816 | 8,057 | 25 | 111.7 | 0.3 |
| From 30–39 | 6,167,587 | 13,580 | 46 | 220.2 | 0.7 |
| From 40–49 | 7,813,183 | 21,221 | 140 | 271.6 | 1.8 |
| From 50–59 | 6,974,007 | 26,461 | 384 | 379.4 | 5.5 |
| From 60–69 | 5,281,870 | 22,721 | 1,099 | 430.2 | 20.8 |
| From 70–79 | 3,900,549 | 21,739 | 3,215 | 557.3 | 82.4 |
| ≥80 | 2,860,952 | 30,415 | 7,403 | 1,063.1 | 258.8 |
| Total | 47,100,398 | 144,911 | 12,314 | 307.7 | 26.1 |

Source: Own elaboration from the Carlos III Health Institute data [23] and the National Institute of Statistics [22]

**Table 3. Percentage change from pre-pandemic mobility and EU+UK lockdown rate (April 17, 2020).**

| Country | Retail and recreation | Grocery and pharmacy | Parks | Transit stations | Workplaces | Residential | Lockdown Index |
|---|---|---|---|---|---|---|---|
| **Sweden** | **-18** | **-3** | **56** | **-36** | **-32** | **11** | **26.0** |
| **Hungary** | **-44** | **-14** | **1** | **-48** | **-45** | **19** | **28.5** |
| **Estonia** | **-46** | **-16** | **-2** | **-46** | **-48** | **20** | **29.7** |
| **Norway** | **-24** | **3** | **68** | **-44** | **-44** | **16** | **33.2** |
| **Lithuania** | **-54** | **-10** | **21** | **-53** | **-50** | **21** | **34.8** |
| **Finland** | **-42** | **-12** | **32** | **-59** | **-48** | **17** | **35.0** |
| **Netherlands** | **-41** | **-10** | **38** | **-59** | **-45** | **17** | **35.0** |
| **Germany** | **-55** | **-4** | **49** | **-49** | **-43** | **16** | **36.0** |
| **North Macedonia** | **-58** | **-2** | **-5** | **-58** | **-70** | **23** | **36.0** |
| **Slovakia** | **-63** | **-12** | **27** | **-53** | **-44** | **18** | **36.2** |
| **Austria** | **-65** | **-16** | **-10** | **-58** | **-52** | **20** | **36.8** |
| **Bosnia and Herzegovina** | **-65** | **-29** | **-3** | **-56** | **-54** | **19** | **37.7** |
| **Croatia** | **-63** | **-27** | **0** | **-67** | **-54** | **21** | **38.7** |
| **Slovenia** | **-68** | **-29** | **-12** | **-55** | **-50** | **23** | **39.5** |
| **Malta** | **-64** | **-21** | **-22** | **-49** | **-56** | **28** | **40.0** |
| **Poland** | **-53** | **-28** | **-44** | **-61** | **-42** | **20** | **41.3** |
| Belgium | -75 | -19 | -14 | -66 | -63 | 30 | 44.5 |
| Denmark | -26 | -6 | 127 | -50 | -45 | 15 | 44.8 |
| Greece | -75 | -2 | -23 | -69 | -73 | 30 | 45.3 |
| Bulgaria | -58 | -19 | -35 | -62 | -73 | 26 | 45.5 |
| Romania | -64 | -21 | -46 | -67 | -68 | 24 | 48.3 |
| Luxembourg | -81 | -20 | -21 | -68 | -71 | 37 | 49.7 |
| United Kingdom | -75 | -30 | -33 | -71 | -68 | 29 | 51.0 |
| Portugal | -69 | -30 | -58 | -73 | -62 | 35 | 54.5 |
| France | -81 | -33 | -62 | -79 | -68 | 35 | 59.7 |
| Italy | -79 | -34 | -75 | -76 | -63 | 32 | 59.8 |
| Spain | -89 | -45 | -77 | -81 | -67 | 33 | 65.3 |
| Average | -59.1 | -18.1 | -4.6 | -59.7 | -55.5 | 23.5 | 42.0 |

Source: Own elaboration based on data from Google [24].

From this index, we calculate the average value of the whole sample of countries, and which takes the value 42. All the countries below this value, marked in bold, make up the reference group of countries with a low level of lockdown, which we then use to calculate the thresholds of lockdown exit strategies in rates of infected and accumulated deaths per 100,000 inhabitants.

## D.- Epidemiological data on COVID-19 in Europe

Table 4 includes the epidemiological data for EU countries plus the UK. We understand that, due to their socioeconomic and health characteristics, they offer an adequate reference group for comparisons to be made with Spain. Data are taken from the 91[st] report on the evolution of COVID-19 issued by the World Health Organization for April 20, 2020. Population data are taken from Eurostat.

## 4. Results

Applying the described methodology, we built the SSMD for the Spanish case for a territorial level of provinces that gives the results shown in Fig 2.

**Table 4. Accumulated COVID-19 infection and death rates per 100,000 inhabitants in the European Union and the United Kingdom on April 20, 2020.** Data by country.

| Country | Population | Confirmed cases | Deaths | Confirmed cases/ 100.000 hab | Deaths / 100.000 hab |
|---|---|---|---|---|---|
| Sweden | 10,183,175 | 14,385 | 1,540 | 141.26 | 15.12 |
| Hungary | 9,768,785 | 1,984 | 199 | 20.31 | 2.04 |
| Estonia | 1,320,884 | 1,528 | 40 | 115.68 | 3.03 |
| Norway | 5,314,336 | 7,068 | 154 | 133.00 | 2.90 |
| Lithuania | 2,789,533 | 1,326 | 36 | 47.53 | 1.29 |
| Finland | 5,518,050 | 3,783 | 102 | 68.56 | 1.85 |
| Netherlands | 17,231,017 | 32,655 | 3,684 | 189.51 | 21.38 |
| Germany | 82,927,922 | 141,672 | 4,404 | 170.84 | 5.31 |
| North Macedonia | 2,077,132 | 1,207 | 51 | 58.11 | 2.46 |
| Slovakia | 5,447,011 | 1,161 | 12 | 21.31 | 0.22 |
| Austria | 8,847,037 | 14,710 | 452 | 166.27 | 5.11 |
| Bosnia and Herzegovina | 3,324,000 | 1,286 | 46 | 38.69 | 1.38 |
| Croatia | 4,089,400 | 1,871 | 47 | 45.75 | 1.15 |
| Slovenia | 2,067,372 | 1,330 | 74 | 64.33 | 3.58 |
| Malta | 483,530 | 427 | 3 | 88.31 | 0.62 |
| Poland | 37,978,548 | 9,287 | 360 | 24.45 | 0.95 |
| Belgium | 11,422,068 | 38,496 | 5,683 | 337.03 | 49.75 |
| Denmark | 5,797,446 | 7,384 | 355 | 127.37 | 6.12 |
| Greece | 10,727,668 | 2,235 | 110 | 20.83 | 1.03 |
| Bulgaria | 7,050,000 | 915 | 43 | 12.98 | 0.61 |
| Romania | 19,473,936 | 8,746 | 434 | 44.91 | 2.23 |
| Luxembourg | 607,728 | 3,550 | 73 | 584.14 | 12.01 |
| United Kingdom | 66,270,000 | 120,071 | 16,060 | 181.18 | 24.23 |
| Portugal | 10,281,762 | 20,206 | 714 | 196.52 | 6.94 |
| France | 66,987,244 | 111,463 | 19,689 | 166.39 | 29.39 |
| Italy | 60,431,283 | 178,972 | 23,660 | 296.16 | 39.15 |
| Spain | 46,723,749 | 195,944 | 20,453 | 419.37 | 43.77 |
| Total | 505,140,616 | 923,662 | 98,478 | 182.85 | 19.50 |

Source: Own elaboration based on data from the World Health Organization [25] and Eurostat [26].

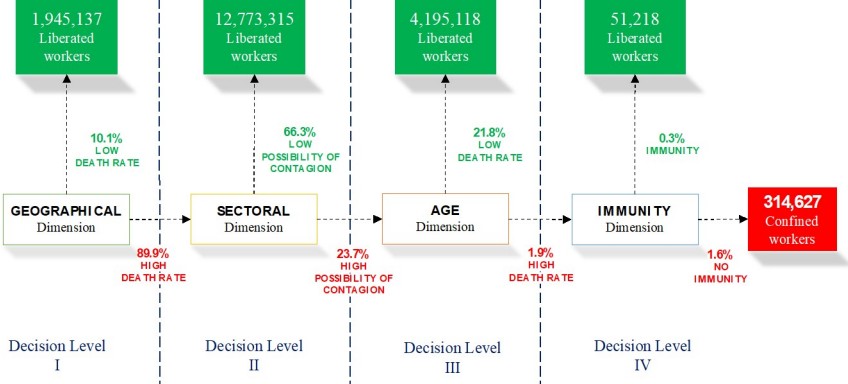

**Fig 2. Lockdown de-escalation strategy proposed for the Spanish case by applying an SSMD in a COVID-19 pandemic situation.** Source: Own elaboration.

For the geographical scope (Decision Level I), the aim is to prevent the spread of COVID-19. We thus calculate the threshold corresponding to the accumulated transmission levels in European countries with a below average degree of lockdown, combining Tables 5 and 6 and obtaining, as a result, a threshold of 118.2 transmissions per 100,000 inhabitants. Provinces whose mortality rates are below the threshold are Almería, Cádiz, Huelva, Murcia, and Las Palmas, which represents 1,945,137 social security affiliates who would be able to return to work. Provinces that fail to meet this requirement are moved to Decision Level II.

As explained in the methodology section, Decision Level II proposes a lockdown exit by sector of activity. Since we lack epidemiological data on the pandemic by sector, we cannot use the threshold level and therefore proceed by taking as low contagion possibility sectors those that were considered as such by the Spanish government, and whose activity was not prohibited, except during the two weeks of total lockdown. Obviously, these sectors must be allowed to return because they operate with a low level of social interaction.

Sectors excluded due to high social interaction would be the following:

- Sector G: Wholesale and retail trade; repair of motor vehicles

- Sector I: Hotels, bars, and restaurants

- Sector R: Arts related activities, recreational and leisure activities

The remaining sectors of activity would be included among those qualified as sectors with a low possibility of contagion, and their workers would return to work. Applying this criterion would affect 12,773,315 of those affiliated to the social security system who could then return to work.

Decision Level III involves assessing the lockdown de-escalation process in sectors of activity considered to be highly contagious in provinces where deconfinement is not full-scale. For this phase, we used the age dimension, given that the disease has a different epidemiological impact, as seen in Table 2. In this case, in order to determine who can return to work we again take data from the European reference group. When going down to the personal level, the fundamental concern is to prevent people who, because of their age may develop lethal COVID-19, from going to work. As a result, we take as a reference the average accumulated mortality rate of the reference group, which is 5.6 per 100,000 people. Applying this criterion would mean that in these sectors workers up to 59 years of age who are not able to work could return to work. Workers aged 60 or over would move to Decision Level IV. The total number of workers returning to their jobs at this stage would be 4,195,118.

Workers who would go on to Decision Level IV would be those aged 60 or over, and who belong to the sectors of activity with a high possibility of contagion in provinces with cumulative contagion rates above the average of the European reference group of countries with a low degree of lockdown exit. These workers would be subject to immunity criteria: serological IgG antibody tests would be carried out, and if they tested positive they could return to work. As already stated, taking as a reference the serological study carried out in Germany [27] on a community with a high incidence of the disease, Gangelt, which could be deemed to resemble Spain, 14% could test positive. This would mean that a further 51,218 workers could return to work as they are immune.

The result of applying these criteria to the available data is that 314,627 workers would be left to return to work after the SSMD. Table 5 shows the relevant figures by province at each Decision Level. The final column includes the provincial distribution of workers who would be waiting to return to work and for whom income replacement measures would be required.

Once we have obtained the number of liberated workers by province and age from the SSMD, we can establish an estimate of the mortality rate associated with this lockdown exit

**Table 5. Lockdown de-escalation strategy proposed for the Spanish case after applying an SSMD in a COVID-19 pandemic situation.** Provincial results expressed in number of social security affiliates.

| Autonomous Communities | Province | SS affiliates | Total Liberated Workers | Liberated Workers Level I | Workers to Level II | Liberated Workers Level II | Workers to Level III | Liberated Workers Level III | Workers to Level IV | Liberated Workers Level IV | Workers Confined |
|---|---|---|---|---|---|---|---|---|---|---|---|
| ANDALUSIA | Almeria | 304,540 | 304,540 | 304,540 | 0 | 0 | 0 | 0 | 0 | 0 | 0 |
| | Cadiz | 375,817 | 375,817 | 375,817 | 0 | 0 | 0 | 0 | 0 | 0 | 0 |
| | Cordoba | 296,800 | 291,382 | 0 | 296,800 | 231,420 | 65,380 | 59,080 | 6,300 | 882 | 5,418 |
| | Granada | 338,485 | 331,396 | 0 | 338,485 | 244,538 | 93,947 | 85,704 | 8,243 | 1,154 | 7,089 |
| | Huelva | 235,290 | 235,290 | 235,290 | 0 | 0 | 0 | 0 | 0 | 0 | 0 |
| | Jaen | 231,507 | 227,852 | 0 | 231,507 | 184,497 | 47,010 | 42,760 | 4,250 | 595 | 3,655 |
| | Malaga | 621,717 | 608,644 | 0 | 621,717 | 417,001 | 204,716 | 189,515 | 15,201 | 2,128 | 13,073 |
| | Sevilla | 744,831 | 733,473 | 0 | 744,831 | 556,988 | 187,843 | 174,636 | 13,207 | 1,849 | 11,358 |
| ARAGON | Huesca | 99,315 | 97,155 | 0 | 99,315 | 74,277 | 25,038 | 22,526 | 2,512 | 352 | 2,160 |
| | Teruel | 54,693 | 53,743 | 0 | 54,693 | 42,753 | 11,940 | 10,835 | 1,105 | 155 | 950 |
| | Zaragoza | 422,608 | 415,354 | 0 | 422,608 | 323,631 | 98,977 | 90,543 | 8,435 | 1,181 | 7,254 |
| ASTURIAS | Asturias | 363,469 | 354,734 | 0 | 363,469 | 262,830 | 100,639 | 90,482 | 10,157 | 1,422 | 8,735 |
| BALEARS, ILLES | Balearic, Islands | 447,918 | 437,949 | 0 | 447,918 | 303,942 | 143,976 | 132,384 | 11,592 | 1,623 | 9,969 |
| CANARY ISLANDS | Palmas, Las | 432,996 | 432,996 | 432,996 | 0 | 0 | 0 | 0 | 0 | 0 | 0 |
| | Santa Cruz de Tenerife | 386,220 | 377,124 | 0 | 386,220 | 236,173 | 150,047 | 139,471 | 10,576 | 1,481 | 9,096 |
| CANTABRIA | Cantabria | 216,443 | 211,750 | 0 | 216,443 | 159,565 | 56,878 | 51,421 | 5,457 | 764 | 4,693 |
| CASTILE—LA MANCHA | Albacete | 140,332 | 137,802 | 0 | 140,332 | 104,964 | 35,368 | 32,426 | 2,942 | 412 | 2,530 |
| | Ciudad Real | 166,369 | 163,635 | 0 | 166,369 | 126,881 | 39,488 | 36,308 | 3,180 | 445 | 2,734 |
| | Cuenca | 76,715 | 75,471 | 0 | 76,715 | 60,010 | 16,705 | 15,259 | 1,446 | 202 | 1,244 |
| | Guadalajara | 90,944 | 89,637 | 0 | 90,944 | 71,110 | 19,834 | 18,314 | 1,520 | 213 | 1,307 |
| | Toledo | 230,813 | 226,971 | 0 | 230,813 | 175,390 | 55,423 | 50,956 | 4,467 | 625 | 3,842 |
| CASTILE AND LEON | Avila | 52,886 | 51,496 | 0 | 52,886 | 39,110 | 13,776 | 12,159 | 1,617 | 226 | 1,390 |
| | Burgos | 147,291 | 144,317 | 0 | 147,291 | 113,768 | 33,523 | 30,065 | 3,458 | 484 | 2,974 |
| | Leon | 157,370 | 153,555 | 0 | 157,370 | 115,220 | 42,150 | 37,714 | 4,436 | 621 | 3,815 |
| | Palencia | 63,551 | 62,186 | 0 | 63,551 | 49,756 | 13,795 | 12,208 | 1,587 | 222 | 1,365 |
| | Salamanca | 119,726 | 116,811 | 0 | 119,726 | 88,064 | 31,662 | 28,273 | 3,389 | 474 | 2,915 |
| | Segovia | 60,837 | 59,388 | 0 | 60,837 | 45,409 | 15,428 | 13,743 | 1,685 | 236 | 1,449 |
| | Soria | 38,958 | 38,229 | 0 | 38,958 | 31,135 | 7,823 | 6,976 | 847 | 119 | 729 |
| | Valladolid | 217,966 | 213,932 | 0 | 217,966 | 166,383 | 51,583 | 46,892 | 4,691 | 657 | 4,034 |
| | Zamora | 56,499 | 54,983 | 0 | 56,499 | 42,249 | 14,250 | 12,487 | 1,763 | 247 | 1,516 |
| CATALONIA | CATALONIA | 3,442,733 | 3,382,830 | 0 | 3,442,733 | 2,521,226 | 921,507 | 851,853 | 69,654 | 9,752 | 59,903 |
| CEUTA | Ceuta | 23,200 | 22,634 | 0 | 23,200 | 16,407 | 6,793 | 6,135 | 658 | 92 | 566 |
| VALENCIAN COMMUNITY | Alicante | 660,665 | 645,509 | 0 | 660,665 | 440,554 | 220,111 | 202,487 | 17,624 | 2,467 | 15,157 |
| | Castellon | 235,799 | 231,557 | 0 | 235,799 | 169,240 | 66,559 | 61,626 | 4,933 | 691 | 4,242 |
| | Valencia | 1,031,398 | 1,011,899 | 0 | 1,031,398 | 729,638 | 301,760 | 279,087 | 22,673 | 3,174 | 19,499 |
| EXTREMADURA | Badajoz | 246,370 | 242,296 | 0 | 246,370 | 190,270 | 56,100 | 51,362 | 4,738 | 663 | 4,074 |
| | Caceres | 141,661 | 138,849 | 0 | 141,661 | 109,797 | 31,864 | 28,594 | 3,270 | 458 | 2,812 |
| GALICIA | GALICIA | 1,012,422 | 991,689 | 0 | 1,012,422 | 746,440 | 265,982 | 241,874 | 24,109 | 3,375 | 20,733 |
| MADRID, COMMUNITY OF | Madrid | 3,279,409 | 3,231,007 | 0 | 3,279,409 | 2,487,826 | 791,583 | 735,301 | 56,282 | 7,879 | 48,402 |
| MELILLA | Melilla | 24,501 | 23,930 | 0 | 24,501 | 17,124 | 7,377 | 6,714 | 663 | 93 | 571 |

*(Continued)*

**Table 5.** (Continued)

| Autonomous Communities | Province | SS affiliates | Total Liberated Workers | Liberated Workers Level I | Workers to Level II | Liberated Workers Level II | Workers to Level III | Liberated Workers Level III | Workers to Level IV | Liberated Workers Level IV | Workers Confined |
|---|---|---|---|---|---|---|---|---|---|---|---|
| MURCIA, REGION OF | Murcia | 596,494 | 596,494 | 596,494 | 0 | 0 | 0 | 0 | 0 | 0 | 0 |
| NAVARRE | Navarra | 288,913 | 284,748 | 0 | 288,913 | 228,042 | 60,871 | 56,028 | 4,843 | 678 | 4,165 |
| BASQUE COUNTRY | Araba/Álava | 159,887 | 157,768 | 0 | 159,887 | 128,022 | 31,865 | 29,401 | 2,464 | 345 | 2,119 |
|  | Gipuzkoa | 325,940 | 320,360 | 0 | 325,940 | 252,783 | 73,157 | 66,669 | 6,488 | 908 | 5,580 |
|  | Vizcaya | 487,401 | 478,362 | 0 | 487,401 | 370,786 | 116,615 | 106,104 | 10,511 | 1,472 | 9,039 |
| RIOJA, LA | Rioja, La | 129,716 | 127,244 | 0 | 129,716 | 98,096 | 31,620 | 28,746 | 2,874 | 402 | 2,472 |
| TOTAL | | 19,279,415 | 18,964,788 | 1,945,137 | 17,334,278 | 12,773,315 | 4,560,963 | 4,195,118 | 365,845 | 51,218 | 314,627 |

Source: Own elaboration.

strategy by applying equation (1). The result we obtain is a mortality rate of 1.35 deaths per 100,000 inhabitants, which marks a low level of mortality associated with the age composition of the workers and the epidemiological incidence of COVID-19 in their age groups. Table 6 shows the estimated deaths by province and the national total for each phase.

## 5. Discussion

The results presented are a simplified approximation to SSMD-type decision making. The authorities, in our case the Spanish Government, clearly have information at a much more dis-aggregated level that would allow the SSMD results to be fine-tuned to a far greater degree.

The SSMD presented poses obvious problems in terms of quantification. We have taken the month of February as a reference, while the Spanish Government is proposing deconfinement for May. By using February, we underestimate the possibilities of improving affiliation to the social security system since May presents higher levels of affiliation due to seasonality. We could have used May 2019 as a reference month, but we considered February to be preferable as this seasonality is associated with the sectors whose activities have been prohibited by the COVID-19 pandemic.

Moreover, we maintain a purely quantitative accounting approach: that is, we do not take into consideration the economic dynamics implicit in the evolution of the economy in the face of a shock such as that triggered by the COVID-19 pandemic. A more precise economic approach requires analysing the foreseeable evolution of the economy and employment by considering supply factors (labour activities) such as teleworking, and demand factors (non-labour activities) such as the effect of self-isolation and social distancing rules that are maintained after the lockdown de-escalation process. The research of Baqaee et al. [10] is relevant to these aspects, as they present a very broad analysis of this type of factor, broken down by sectors, with forecasts for the evolution of the COVID-19 pandemic in the USA depending on the type of deconfinement strategy and social distancing (non-labour) in place after reopening, a fundamental factor in how successful the fight against the pandemic proves to be. Decision-making based on an SSMD model will be all the more effective the greater the amount of economic and epidemiological criteria that are analysed and taken into account.

There are several advantages to the SSMD lockdown exit strategy. On the one hand, it offers us a target-oriented, rather than tailor-made, view of the problem. Logical reasoning would initially lead us to consider that equal measures give rise to equal results. However, nothing could be further from the truth when it is scientifically evaluated. A country succeeds in

**Table 6. Mortality per 100,000 workers associated with the lockdown exit strategy according to the SSMD.** Estimate by province according to the age pyramid.

| Autonomous Communities | Province | Deaths Level I / 100.000 hab | Deaths Level II / 100.000 hab | Deaths Level III / 100.000 hab | Deaths Level IV / 100.000 hab | Deaths / 100.000 hab |
|---|---|---|---|---|---|---|
| ANDALUSIA | Almeria | 1.48 | 0.00 | 0.00 | 0.00 | 1.48 |
| | Cadiz | 1.10 | 0.00 | 0.00 | 0.00 | 1.10 |
| | Cordoba | 0.00 | 1.23 | 0.16 | 0.00 | 1.39 |
| | Granada | 0.00 | 1.05 | 0.20 | 0.00 | 1.24 |
| | Huelva | 1.52 | 0.00 | 0.00 | 0.00 | 1.52 |
| | Jaen | 0.00 | 1.18 | 0.15 | 0.00 | 1.33 |
| | Malaga | 0.00 | 0.90 | 0.23 | 0.00 | 1.13 |
| | Sevilla | 0.00 | 1.00 | 0.18 | 0.00 | 1.19 |
| ARAGON | Huesca | 0.00 | 1.43 | 0.22 | 0.00 | 1.65 |
| | Teruel | 0.00 | 1.32 | 0.18 | 0.00 | 1.50 |
| | Zaragoza | 0.00 | 1.31 | 0.20 | 0.00 | 1.51 |
| ASTURIAS | Asturias | 0.00 | 1.11 | 0.20 | 0.00 | 1.30 |
| BALEARS, ILLES | Balearic, Islands | 0.00 | 0.98 | 0.23 | 0.00 | 1.21 |
| CANARY ISLANDS | Palmas, Las | 1.36 | 0.00 | 0.00 | 0.00 | 1.36 |
| | Santa Cruz de Tenerife | 0.00 | 0.81 | 0.28 | 0.00 | 1.09 |
| CANTABRIA | Cantabria | 0.00 | 1.14 | 0.19 | 0.00 | 1.33 |
| CASTILE—LA MANCHA | Albacete | 0.00 | 1.05 | 0.18 | 0.00 | 1.23 |
| | Ciudad Real | 0.00 | 0.97 | 0.15 | 0.00 | 1.12 |
| | Cuenca | 0.00 | 1.22 | 0.17 | 0.00 | 1.39 |
| | Guadalajara | 0.00 | 1.02 | 0.15 | 0.00 | 1.17 |
| | Toledo | 0.00 | 0.95 | 0.15 | 0.00 | 1.11 |
| CASTILE AND LEON | Avila | 0.00 | 1.18 | 0.18 | 0.00 | 1.35 |
| | Burgos | 0.00 | 1.40 | 0.19 | 0.00 | 1.58 |
| | Leon | 0.00 | 1.11 | 0.18 | 0.00 | 1.29 |
| | Palencia | 0.00 | 1.43 | 0.17 | 0.00 | 1.60 |
| | Salamanca | 0.00 | 1.19 | 0.19 | 0.00 | 1.38 |
| | Segovia | 0.00 | 1.35 | 0.20 | 0.00 | 1.55 |
| | Soria | 0.00 | 1.59 | 0.18 | 0.00 | 1.77 |
| | Valladolid | 0.00 | 1.30 | 0.20 | 0.00 | 1.50 |
| | Zamora | 0.00 | 1.20 | 0.17 | 0.00 | 1.37 |
| CATALONIA | CATALONIA | 0.00 | 1.18 | 0.22 | 0.00 | 1.40 |
| CEUTA | Ceuta | 0.00 | 0.81 | 0.16 | 0.00 | 0.96 |
| VALENCIAN COMMUNITY | Alicante | 0.00 | 0.89 | 0.23 | 0.00 | 1.12 |
| | Castellon | 0.00 | 1.07 | 0.23 | 0.00 | 1.29 |
| | Valencia | 0.00 | 1.04 | 0.23 | 0.00 | 1.26 |
| EXTREMADURA | Badajoz | 0.00 | 1.10 | 0.16 | 0.00 | 1.26 |
| | Caceres | 0.00 | 1.22 | 0.16 | 0.00 | 1.39 |
| GALICIA | GALICIA | 0.00 | 1.11 | 0.19 | 0.00 | 1.31 |
| MADRID, COMMUNITY OF | Madrid | 0.00 | 1.29 | 0.22 | 0.00 | 1.51 |
| MELILLA | Melilla | 0.00 | 0.77 | 0.16 | 0.00 | 0.93 |
| MURCIA, REGION OF | Murcia | 1.40 | 0.00 | 0.00 | 0.00 | 1.40 |
| NAVARRE | Navarra | 0.00 | 1.32 | 0.18 | 0.00 | 1.51 |
| BASQUE COUNTRY | Araba/Álava | 0.00 | 1.47 | 0.19 | 0.00 | 1.66 |

*(Continued)*

**Table 6.** (Continued)

| Autonomous Communities | Province | Deaths Level I / 100.000 hab | Deaths Level II / 100.000 hab | Deaths Level III / 100.000 hab | Deaths Level IV / 100.000 hab | Deaths / 100.000 hab |
|---|---|---|---|---|---|---|
| | Gipuzkoa | 0.00 | 1.42 | 0.20 | 0.00 | 1.62 |
| | Vizcaya | 0.00 | 1.33 | 0.21 | 0.00 | 1.53 |
| RIOJA, LA | Rioja, La | 0.00 | 1.25 | 0.20 | 0.00 | 1.45 |
| TOTAL | | 0.15 | 1.02 | 0.19 | 0.00 | 1.35 |

Source: Own elaboration from Tables 2 and 3.

reducing daily infections of COVID-19 by lockdown strategies. Yet in areas of the country where no infection have occurred it has failed to reduce anything. Isolating those infected, social distancing, frequent hand-washing, wearing a face mask, avoiding concentrations of people in confined places, preventing long distance travel, etc., are measures that have a similar effect in any area, while confining an entire population to their homes can have a negligible effect in a rural area but a very significant one in a large city like Madrid or Barcelona. The same measures give different results depending on the initial conditions and the environment. From our point of view, a lockdown de-escalation strategy must aim to achieve the same results, but not to apply the same measures. This should be so because we must not incur excessive and avoidable economic costs. The strategy proposed here adapts to this way of looking at the problem, and is closer to the viewpoint of the person who must make the decision.

Another advantage of the strategy we propose is that it makes it possible to substantially reduce the populations targeted by urgent public health measures. In the study, we identified, for economic purposes, as an urgent target population for serological tests those workers aged 60 in the wholesale and retail trade and repair of motor vehicles; hotels, bars, and restaurants; arts related activities, recreational and leisure activities sectors. In other words, we reduce the immediate need for serological testing to 1.9% of workers. This relieves the authorities from the pressure of having to get more tests than are immediately required and it also means the testing procedure can be better organised. The WHO already applied this approach when it recommended that full-scale testing of health-care workers should be a priority.

Furthermore, by significantly reducing the affected population by applying geographical and sectoral criteria, we reduce all the moral problems associated with age-selective lockdown and the need for an immunity passport, which would also be greatly reduced. We also curb the public costs derived from having to cover the income of furloughed employees by positing a high level of lockdown exit measures and by reducing the number of people affected.

Finally, the proposal is very versatile as it allows for more or less strict public health criteria to be established, which differ from those being introduced here. Public authorities can use other criteria that are more in line with public health needs at any given time without invalidating the method. The flexibility of the strategy is also feasible at other levels, as we can disaggregate the geographical, sectorial, or age level, while maintaining the structure of the decision-making system. Disaggregation helps to refine the result, improve decision-making and reduce health risks.

## 6. Conclusions

The current health pandemic has become a social and public health crisis that is unprecedented in our recent history. From the point of view of public health and social justice, understood from a Rawls and Sen perspective, the first obligation is to save lives and to support all health workers as well as all of those groups that ensure our day to day existence. However,

although health is the most urgent issue, the resources dedicated to health protection are related to economic development, and post-pandemic economic needs may entail enormous social and personal costs. At present, the models developed considering different contingency scenarios indicate that Spaniards could lose around 3,602 euros per year in terms of lower GDP per capita [28–32]. In other words, the most plausible forecasts for the coming months suggest there will be sharp falls in GDP and increases in public deficit as needs grow. These same consequences can be extended to the rest of the world [9, 33], where countries will be faced with a trade-off between health and economy that must be resolved through the lockdown de-escalation strategy.

The work presented here offers a method for structuring this decision for a lockdown de-escalation strategy in the countries affected by the COVID-19 pandemic. To do so, we define a Sequential Selective Multidimensional Decisioning (SSMD) process based on four dimensions (geographical area, sector of activity, age, and immunity) ordered sequentially. In each of these dimensions, a decision is made as to which workers may return to work, considering the epidemiological characteristics of the country, in our case Spain, and of the reference group of European countries with low levels of lockdown. Once the strategy has been defined, we quantitatively calculate the incidence of the lockdown exit strategy for Spain, based on affiliation to the social security system prior to the pandemic.

Specifically, we conclude that a lockdown de-escalation process involving 98.55% of those affiliated to the social security system in Spain at the end of February 2020 is feasible without putting at risk in the workplace the population most likely to be affected by COVID-19. This is, however, conditional upon ensuring safety, health and social distancing in those workplaces, guaranteeing that the number of tests carried out is increased and that an adequate traceability of the network of contacts of cases that do test positive for COVID-19 is established.

Finally, another fundamental contribution of this work is that the SSMD also makes it possible to determine the working population targeted by the serological IgG antibody tests and to evaluate the economic measures needed to replace the income of those affected.

Given the characteristics of the COVID-19 pandemic, which concentrates the highest mortality rates in non-working ages, the results represent a modest improvement over a broad reopening, since only 1.63% of workers remain in lockdown and the ultimate effectiveness of the fight against the pandemic depends fundamentally on "strong restrictions on non-work social contacts". However, this does not detract from the usefulness of the SSMD as a method for "smart" reopening, since its utility depends on the pyramid of incidence, the mortality of the pandemic in question, and the effectiveness of non-work behaviour vis-à-vis the pandemic.

## Supporting information

**S1 Appendix. Notes about the situation in Spain.**
(DOCX)

**S2 Appendix. Affiliated workers by province.**
(DOCX)

## Author Contributions

**Conceptualization:** Luis Angel Hierro, David Cantarero, David Patiño, Daniel Rodríguez-Pérez de Arenaza.

**Data curation:** Luis Angel Hierro, David Cantarero, David Patiño, Daniel Rodríguez-Pérez de Arenaza.

**Investigation:** Luis Angel Hierro, David Cantarero, David Patiño, Daniel Rodríguez-Pérez de Arenaza.

**Methodology:** Luis Angel Hierro, David Cantarero, David Patiño, Daniel Rodríguez-Pérez de Arenaza.

**Supervision:** Luis Angel Hierro, David Cantarero, David Patiño, Daniel Rodríguez-Pérez de Arenaza.

**Writing – original draft:** Luis Angel Hierro, David Cantarero, David Patiño, Daniel Rodríguez-Pérez de Arenaza.

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
