## [Decision Letter · Decision Letter 0]

9 Jul 2020

PONE-D-20-16646

Who can go back to work when the COVID-19 pandemic remits?

PLOS ONE

Dear Dr. Patino-Rodriguez,

Thank you for submitting your manuscript to PLOS ONE. After careful consideration, we feel that it has merit but does not fully meet PLOS ONE’s publication criteria as it currently stands. Therefore, we invite you to submit a revised version of the manuscript that addresses the points raised during the review process.

Your manuscript has been well-valued by an acknowledged reviewer in the field addressed in the study. However, the referee asks for some further revisions, about which you can find more information in the comments appended below. Particularly, these comments are related to other models and empirical researches previously performed by similar studies, as a manner to optimize and support your manuscript.

We look forward to receiving your revised manuscript.

Kind regards,

Sergio A. Useche, Ph.D.

Academic Editor

PLOS ONE

Journal Requirements:

Reviewers' comments:

Reviewer's Responses to Questions

**Comments to the Author**

1. Is the manuscript technically sound, and do the data support the conclusions?

Reviewer #1: Yes

2. Has the statistical analysis been performed appropriately and rigorously? 

Reviewer #1: Yes

3. Have the authors made all data underlying the findings in their manuscript fully available?

Reviewer #1: Yes

4. Is the manuscript presented in an intelligible fashion and written in standard English?

Reviewer #1: Yes

5. Review Comments to the Author

Reviewer #1: This article is about smart reopening in Spain. As many researchers are doing they are using economic and epidemiology simulations. They should refer to and contrast theirs to ta standard one used in the US.

Reopening Scenarios by Baqaee, Farhi, Mina, Stock, (NBER 27244 May 2020) A complicated model using data on individual actions and non-pharmaceutical interventions in the weeks ending March 8 – May 16, 2020 finds a decision-maker (governor) following reopening guidelines, with information on personal proximity and ability to work from home by sector, make it possible to construct a GDP-to-Risk index of which sectors provide the greatest increment in GDP per marginal increase in R0. – a classic economic tradeoff, Conclusions - a strong economic reopening is possible; a “smart” reopening, preferencing some sectors over others, makes only modest improvements over a broad reopening. All depend on retaining strong restrictions on non-work social contacts. “If non-work contacts – going to bars, shopping without social distancing and masks, large group gatherings, etc. – return only half-way to the pre-COVID-19 baseline and a second wave.”

6. PLOS authors have the option to publish the peer review history of their article (what does this mean?). If published, this will include your full peer review and any attached files.

Reviewer #1: No

---

## [Author Response · Author response to Decision Letter 0]

22 Jul 2020

Dear PLOS ONE reviewer,

We are extremely grateful for your comments on our article “Who can go back to work when the COVID-19 pandemic subsides?”. Please take this letter as an explanation of the changes that we have sought to implement in our work, considering all your suggestions.

We have carefully read the comments made in your report and we have considered how to include your suggestions. We agree with you on the need to highlight the document that you have indicated to us, and we have done so. In this regard, we wished to point out that our article was first completed on a date when the Baqaee, Farhi, Mina and Stock research was not yet accessible (our document was published on May 10 in a preprint version).

Likewise, as you point out, in the version of our article that we are sending now we have included a reflection on the limitation that contagion in non-work activities implies developing a lockdown de-escalation strategy (P.17). The new version of the article that we are sending now also includes the results of the research of Baqaee et al. (2020) (P. 15).

Based on the above, we hope that this new revised version of our article successfully addresses the shortcomings you pointed out in the previous version and that it now meets the requirements for publication in the journal. Please do not hesitate to contact us should you wish to clarify any other question related to our work.

One again, we are grateful for your comments, as these have enabled us to substantially improve our article.

Kind regards.

The authors.

---

## [Editor Report · Decision Letter 1]

14 Aug 2020

Who can go back to work when the COVID-19 pandemic remits?

PONE-D-20-16646R1

Dear Dr. Patino-Rodriguez,

We’re pleased to inform you that your manuscript has been judged scientifically suitable for publication and will be formally accepted for publication once it meets all outstanding technical requirements.

Kind regards,

Sergio A. Useche, Ph.D.

Academic Editor

PLOS ONE